# Assessing the Optimal Regimen: A Systematic Review and Network Meta-Analysis of the Efficacy and Safety of Long-Acting Granulocyte Colony-Stimulating Factors in Patients with Breast Cancer

**DOI:** 10.3390/cancers15143675

**Published:** 2023-07-19

**Authors:** Zhixuan You, Haotian Zhang, Yining Huang, Lei Zhao, Hengjia Tu, Yuzhuo Zhang, Xinqing Lin, Wenhua Liang

**Affiliations:** 1Department of Clinical Medicine, Nanshan School, Guangzhou Medical University, Guangzhou 510182, China; 2019111180@stu.gzhmu.edu.cn (Z.Y.); ttgoodluck0424@163.com (H.Z.); huangyining2019@163.com (Y.H.); 2020111409@stu.gzhmu.edu.cn (L.Z.);; 2Pulmonary and Critical Care Medicine, Guangzhou Institute of Respiratory Health, National Clinical Research Center for Respiratory Disease, National Center for Respiratory Medicine, State Key Laboratory of Respiratory Diseases, The First Affiliated Hospital of Guangzhou Medical University, Guangzhou 510120, China

**Keywords:** granulocyte colony-stimulating factor (G-CSF), pegylated filgrastim (pegfilgratim), biosimilars, drug based, dose based, network meta-analysis, breast cancer

## Abstract

**Simple Summary:**

Patients with breast cancer undergoing chemotherapy are susceptible to prolonged and severe neutropenia. Multiple biosimilar long-acting granulocyte colony-stimulating factors (LA-G-CSFs) have been developed to treat chemotherapy-induced neutropenia. However, which LA-G-CSF is optimal remains inconclusive. Moreover, there is a lack of evidence supporting clinical decisions on LA-G-CSF dose escalation in poor conditions. This systematic review and network meta-analysis aimed to identify the most effective and safe LA-G-CSF regimen and explore the feasibility of dose escalation. Our findings reveal that certain LA-G-CSFs significantly outperform others (including the guideline-recommended pegfilgrastim 6 mg). An increased LA-G-CSF dosage might enhance efficacy without additional severe adverse events. Our research offers essential insights into precise drug selection and personalized dosing in preventing post-chemotherapy neutropenia. However, further high-quality investigations into the efficacy and safety of the use of LA-G-CSFs are warranted in the future.

**Abstract:**

Patients with breast cancer undergoing chemotherapy are susceptible to prolonged and severe neutropenia. Multiple biosimilars of long-acting granulocyte colony-stimulating factors (LA-G-CSFs) have been newly developed to prevent this disease. Nonetheless, which LA-G-CSF regimen has the optimal balance of efficacy and safety remains controversial. Moreover, there is a lack of evidence supporting clinical decisions on LA-G-CSF dose escalation in poor conditions. PubMed, Embase, Cochrane Library, Web of Science, and several Chinese databases were searched (December 2022) to collect randomized controlled trials (RCTs) about LA-G-CSFs preventing chemotherapy-induced neutropenia in breast cancer patients. No restrictions were imposed on language. A Bayesian network meta-analysis was performed. We assessed the incidence of severe neutropenia (SN) and febrile neutropenia (FN), the duration of SN (DSN), and the absolute neutrophil account recovery time (ANCrt) for efficacy, while the incidence of severe adverse events (SAE) was assessed for safety. The study was registered in PROSPERO (CRD42022361606). A total of 33 RCTs were included. Our network meta-analysis demonstrated that lipegfilgrastim 6 mg and eflapegrastim 13.2 mg outperformed other LA-G-CSFs with high efficacy rates and few safety concerns (SUCRA of lipegfilgrastim 6 mg: ANC rt 95.2%, FN 97.4%; eflapegrastim 13.2 mg: FN 87%, SN 89.3%). Additionally, 3.6 mg, 4.5 mg, 6 mg, and 13.2 mg dosages all performed significantly better than 1.8 mg in reducing the duration of SN (3.6 mg: DSN, SMD −0.68 [−1.13, −0.22; moderate]; 4.5 mg: −0.87 [−1.57, −0.17; low]; 6 mg: −0.89 [−1.49, −0.29; moderate]; 13.2 mg: −1.02 [1.63, −0.41; high]). Increasing the dosage from the guideline-recommended 6 mg to 13.2 mg can reduce both the duration and incidence of SN (SMD −0.13 [−0.24 to −0.03], RR 0.65 [0.43 to 0.96], respectively), with no significant difference in SAE. For patients with breast cancer, lipegfilgrastim 6 mg and eflapegrastim 13.2 mg might be the most effective regimen among LA-G-CSFs. Higher doses of LA-G-CSF may enhance efficacy without causing additional SAEs.

## 1. Introduction

Breast cancer has become the most prevalent malignancy worldwide and caused more than 680,000 deaths in 2020 [1]. Almost all breast cancer patients need immunosuppressive chemotherapy [2], which features long treatment cycles and high doses of drugs in solid cancers. Febrile neutropenia (FN) and severe neutropenia (SN) are severe complications that commonly occur in breast cancer patients receiving chemotherapy [3]. Persistent SN and FN often lead to treatment delays, dosage reduction, and an increased risk of infection, ultimately resulting in poor treatment outcomes, with mortality rates of 6.8% and 9.5%, respectively [4]. Consequently, reducing the occurrence and duration of these adverse events would be of exceptional importance for improving treatment outcomes and quality of life for breast cancer patients.

Granulocyte colony-stimulating factor (G-CSF) is a conventional intervention for preventing chemotherapy-induced neutropenia [5,6]. Currently, various G-CSF drugs, such as short-acting G-CSF (SA-G-CSF), long-acting G-CSF (LA-G-CSF), and their analogues, are available for patients undergoing chemotherapy [7,8]. The efficacy of LA-G-CSF is reported to be comparable to that of SA-G-CSF [9]. However, recent randomized controlled trials (RCTs) have suggested that certain LA-G-CSFs would be more effective for reducing the incidence of FN and SN, and shortening the duration of SN (DSN) [10,11,12]. Although several systematic reviews have compared the efficacy and safety of LA-G-CSFs and SA-G-CSF, they did not employ a network meta-analysis methodology and had varying criteria for study inclusion, which compromised the quality of their evidence. Moreover, these reviews did not include the G-CSF medications that are widely utilized in China. Additionally, the conclusions drawn from these reviews are inconsistent. Hence, differences in the efficacy and safety between LA-G-CSF and SA-G-CSF remain controversial.

Direct or indirect comparisons among various LA-G-CSFs are lacking, possibly due to limitations in the study design and feasibility of RCTs. Conducting a pairwise comparison of complex interventions is challenging. For example, Jin You Li, an LA-G-CSF commonly used in China, has never been directly compared with other counterparts used in other nations [13]. Additionally, divergent doses of an LA-G-CSF agent would also result in varying degrees of effectiveness and side effects. A 6 mg dose of pegfilgrastim is recommended by guidelines, but such a dose is supported solely by one RCT with a small sample size [9]. For patients with breast cancer undergoing high-dose and long-cycle chemotherapy regimens, individualized G-CSF protocols and adaptable dosage are vital for optimizing the overall outcomes. However, the optimal LA-G-CSF regimen and dosage remain controversial. A network meta-analysis can address this controversy by comparing various treatments and evaluating their relative effects from multiple perspectives.

Therefore, we conducted this network meta-analysis to assess and compare the efficacy and safety of LA-G-CSFs through drug-based and dose-based analyses, aiming to identify the most effective LA-G-CSF regimens and dosage in the hope of providing a reliable reference for clinical decision making.

## 2. Method

This study was conducted and reported in accordance with the preferred reporting items for systematic reviews and network meta-analyses (PRISMA-NMA, Appendix A) [14]. The study protocol was pre-registered on PROSPERO (registration No. CRD42022361606).

### 2.1. Search Strategy and Selection Criteria

We searched the following electronic databases from the inception of the databases to 1 December 2022: PubMed, Embase, Cochrane Library, and Web of Science. We also searched several prominent Chinese databases: CNKI, Wanfang Database, China Science and Technology Journal Database, and National Science and Technology Library. No restrictions were imposed on language. We contacted the researchers to obtain information about unpublished trials. Detailed search strategies are provided in Appendix A. 

The network meta-analysis incorporated studies that satisfied the following inclusion criteria: (1) participants: patients with breast cancer; (2) intervention: chemotherapy based on LA-G-CSF; (3) control: LA-G-CSF with different doses, SA-G-CSF, or placebo; (4) outcomes: both efficacy and safety outcomes; (5) study design: RCTs; non-RCTs, observational studies, and single-arm trials were excluded; (6) other criteria: when results of the same trial were published in more than one paper, only the latest and/or most complete article was included in the analysis.

### 2.2. Data Extraction and Quality Assessment

Study selection, data extraction, and quality assessment were conducted independently by three investigators (HT, YN, and ZX). Disagreements were settled through discussion. The extracted data mainly contained the first author’s name and year of publication, country, study design, patient characteristics, and chemotherapeutic regimen. The efficacy was assessed based on the incidence of FN and SN, duration of SN, and absolute neutrophil count (ANC) recovery time in chemotherapy cycle 1, while the safety was assessed by the incidence of grade 3–4 adverse events (SAEs) in all treatment cycles. All the extracted data were summarized in a structured table to ensure consistency. For two or more articles that reported the same trial, the latest published/or the most comprehensive article was included in the analysis. The methodological quality of the included studies was evaluated using the Cochrane Risk of Bias (RoB) tool (Odense, Denmark) [15]. 

### 2.3. Outcomes

Efficacy was assessed according to the incidence of FN and SN, duration of SN, and ANC recovery time in CT cycle 1. The European Medicines Agency describes DSN as an acceptable surrogate endpoint for exploratory and confirmatory trials of G-CSF. Chemotherapy-induced neutropenia is a common and serious hematologic toxicity, which not only increases the risk of infection-related mortality but also limits the dose of chemotherapy [16]. SN was defined as an ANC less than 0.5 × 10^9^/L and DSN was defined as the time between the confirmation and disappearance of ANC. FN was typically defined as an oral temperature less than 38.5 °C or 2 consecutive temperatures of over 38.0 °C for 2 h, in conjunction with an ANC less than 0.5 × 10^9^/L (or expected to fall below 0.5 × 10^9^/L). The safety endpoint was the incidence of grade 3–4 AEs across all the treatment cycles, which were evaluated according to the Common Terminology Criteria for Adverse Events (CTCAE), Version 5.0 [17]. The AEs included, but were not limited to, nausea, asthenia, bone pain, diarrhea, fatigue, vomiting, headache, myalgia, infections, and FN.

### 2.4. Data Synthesis and Statistical Analysis

The data synthesis and statistical analysis were conducted from two distinct perspectives: a detailed analysis of each specific drug-related event, and dose-based network meta-analysis. Dichotomous data were expressed as relative ratios (RR), and continuous data were reported as standardized mean differences (SMD, Cohen’s d). The 95% confidence interval (95% CI) of each effect size was provided.

Our analysis included both dose-based and drug-based network meta-analyses, which were conducted using a Bayesian framework and Markov Chain Monte Carlo method (MCMC) to integrate direct and indirect evidence [18,19,20]. The analysis was performed utilizing R version 4.1.3, and the major package used was “gemtc” version 1.0–1. Initially, we built a network using the “mtc.network” function, and then constructed a model using the “mtc.model” function. The Bayesian analysis was conducted using the “mtc.run” function. In the “mtc.model” function, we set the likelihood/link as “binom/log” or “normal/identity” to calculate the log risk ratio (logRR) or the SMD. These models were estimated using “rjags” package. A league table was produced to rank the efficacy of the interventions, and the “exp” function was applied to calculate RR from LogRR. Additionally, forest plots were generated to reveal the comparative effects of different interventions. The surface under the cumulative ranking curve (SUCRA) analysis was performed to present the probability ranking results [18]. The value of SUCRA ranges from 0% (indicating that the intervention always ranks last) to 100% (indicating that the intervention always ranks first). Potential scale reduction factor (PSRF) values were calculated to assess the convergence of the iteration. In particular, a PSRF value of less than 1.2 indicated acceptable model convergence. 

Achieving outstanding consistency between direct and indirect results is crucial to ensure robust results. To appraise inconsistency, we employed a three-step method. Firstly, we compared the fit of the consistency model with that of the inconsistency model. Secondly, the node-split method was utilized to test the consistency assumption, and if *p* < 0.05, inconsistency was deemed significant. Thirdly, the “mtc.anohe” test was performed to assess the homogeneity assumption, with *I*^2^ > 50% indicating significant heterogeneity. If significant heterogeneity was found, we utilized the random-effects model; otherwise, the fixed-effects model was used. The indirect results obtained from the network meta-analysis were compared with the direct results from pairwise meta-analysis to identify the source of inconsistency.

The quality of the evidence of the network meta-analysis was assessed using Confidence in Network Meta-Analysis (CINeMA, sourced from Bern, Switzerland). The “netmeta” package in R was also adopted to perform the network meta-analysis. In addition to grading the confidence of the analysis results, the results of the network meta-analysis using CINeMA were compared with the results obtained from the “gemtc” package.

## 3. Result

A total of 1921 articles were retrieved (Figure 1). After the titles and abstracts were screened to exclude irrelevant studies, the full texts of the remaining 132 articles were read, and 99 were further excluded. Finally, 33 eligible RCTs were included [10,11,12,13,21,22,23,24,25,26,27,28,29,30,31,32,33,34,35,36,37,38,39,40,41,42,43,44,45,46,47,48,49]. Among the included RCTs, 15 were multi-centric. A total of 7988 breast cancer patients were included (Table 1). The timing of G-CSF administration varied among the studies: 6 studies started G-CSF administration 48 h after chemotherapy, whereas the other 27 studies started 24 h after chemotherapy. The most frequently used chemotherapeutic regimen was TAC (39%), followed by AT (36%).

Drug-based (Figure 2A) and dosed-based (Figure 2B) analyses were conducted. In drug-based analyses, 31 interventions were compared. DSN was reported in 25 studies (29 interventions, *n* = 5562), ANC recovery time in 15 studies (18 interventions, *n* = 2980), the incidence of FN in 22 studies (21 interventions, *n* = 5453), and the incidence of SN in 19 studies (23 interventions, *n* = 4819). Moreover, 17 studies were included in the safety assessment (20 interventions, *n* = 3125).

A total of 16 interventions were compared in the dose-based analysis. The dosage of LA-G-CSF ranged from 1.8 mg to 13.2 mg and that of non-pegylated LA-G-CSF (nonPEG-LA-G-CSF) ranged from 30 mg to 50 mg. DSN was reported in 19 studies (14 interventions, *n* = 3689), the incidence FN in 12 studies (13 interventions, *n* = 2302), the incidence of SN in 4 studies (14 interventions, *n* = 3440), and ANC recovery time in 10 trials (12 interventions, *n* = 1581). Nine studies (eight interventions, *n* = 1611) were included in the safety assessment.

### 3.1. Quality Assessment

The results of the quality assessments of the included studies are illustrated in Appendix A. Owing to the severity of the evaluated clinical condition, all trials were open-label. Most of the studies scored positively on “randomization”. However, 7 studies (21%) did not provide detailed descriptions of the randomization process, and 20 studies (60%) showed an unclear RoB in allocation concealment because the studies did not give enough details. Studies were mostly considered to have a low RoB in the blinding of outcome assessments since the reported outcomes for efficacy and safety were mainly objective. Fourteen studies (42%) were graded as having moderate to high RoB regarding incomplete outcome data, due to missing data or the failure to report the reason for them. Seven studies (21%) declared to be funded by pharmaceutical corporations.

### 3.2. Outcome

Ten networks were constructed for efficacy and safety outcomes, five of which were based on the types of drugs and five based on the dosage. No significant heterogeneity was observed in most of the outcomes, except DSN (dose-based) *(I*^2^ = 87%). The sensitivity analysis showed that the existing heterogeneity had no significant impact on the results. According to inconsistency and node-splitting analysis, no significant differences were found in the results of the direct and indirect comparisons in the majority of studies (*p* > 0.05). A PSRF value of 1 confirmed the model’s convergence and the stability of the results. The results of the homogeneity and consistency analysis of each outcome for both drug-based and dose-based analyses are presented in Appendix A.

#### 3.2.1. Some LA-G-CSFs Are Statically More Effective Than SA-G-CSF

Figure 3 shows the pairwise forest plot for the direct and indirect comparisons between the interventions versus control measures of drug-based network meta-analysis for the efficacy outcomes. 

The results of the network meta-analysis indicated differences in these outcomes between the agents, particularly in DSN, as shown in Figure 3A. Empegfilgrastim 6 mg (SMD −0.82 [95% CI, −1.41 to −0.24; CINeMA ranking: high]), empegfilgrastim 7.5 mg (−0.94 [−1.49 to −0.39; high]), mecapegfilgrastim 100 μg/kg (−1 [−1.5 to −0.5; high]), mecapegfilgrastim 6 mg (−0.83 [−1.49 to −0.17; high]), and pegfilgrastim 100 μg/kg (−0.35 [−0.55 to −0.16; high]) were significantly more effective than SA-G-CSF.

Moreover, lipegfilgrastim 6 mg, mecapegfilgrastim 6 mg, eflapegrastim 13.2 mg, LA-EP2006 6 mg, and pegfilgrastim 6 mg had a lower incidence of FN than SA-G-CSF (Figure 3B), with an RR ranging from 0 to 0.74 (very low to high certainty of the evidence). The removal of studies with a high RoB did not change the result (Figure 3C). 

Compared with SA-G-CSF, patients receiving eflapegrastim 13.2 mg (RR 0.62 [95% CI, 0.40 to 0.95; very low]), mecapegfilgrastim (100 µg/kg) (RR 0.76 [0.60 to 0.95; low]), mecapegfilgrastim 6 mg (RR 0.77 [0.61 to 0.96; low]), or lipegfilgrastim 6 mg (RR 0.77 [0.57 to 1.0; moderate]) had a lower incidence of SN (Appendix A). As for other outcomes, patients receiving lipegfilgrastim 6 mg (SMD −1.5 [−2.8 to −0.34; high]) and Xinruibai 100 µg/kg (−1.6 [−2.9 to −0.24; high]) had a shorter ANC recovery time (Appendix A). Patients receiving Shenlida (6 mg, 100 µg/kg) had a lower incidence of grade 3–4 SAEs (6 mg: 0.094 [0.0035 to 0.59; low], 100 µg/kg: 0.23 [0.03 to 0.95; low], Appendix A).

#### 3.2.2. Varying Efficacy of Different LA-G-CSF Agents 

A probability ranking for all five outcomes was produced. Figure 4 and Appendix A shows a visual summary of the final SUCRA analysis results.

The probability ranking showed that mecapegfilgrastim 100 μg/kg, empegfilgrastim 7.5 mg, and empegfilgrastim 6 mg would be the top three most effective in reducing DSN, with the probabilities of 95.9%, 94.8%, and 92.1%, respectively. When compared to the guideline-recommended pegfilgrastim 6 mg, the alternative LA-G-CSF doses and drugs, specifically empegfilgrastim 7.5 mg (SMD −1.8 [95% CrI, −1.8 to −0.45; high]), mecapegfilgrastim 100 μg/kg (SMD −1.2 [95% CrI, −1.8 to −0.55; high]), mecapegfilgrastim 6 mg (SMD −1.0 [95% CrI, −1.8 to −0.26; high]), and empegfilgrastim 6 mg (SMD −1.0 [95% CrI, −1.7 to −0.31; high]), were found to be associated with a statistically significant decrease in DSN (Appendix A). 

Lipegfilgrastim 6 mg (SUCRA 97.4%), mecapegfilgrastim 6 mg (SUCRA 96.8%), and eflapegrastim 13.2 mg (SUCRA 87.0%) may be the top three most effective interventions for reducing the incidence of FN. Eflapegrastim 13.2 mg had the highest probability to become the optimal agent for reducing the incidence of SN compared with other interventions, with a SUCRA value of 89.3%. Except for Xinruibai 100 µg/kg, administration of lipegfilgrastim 6 mg was associated with a shorter ANC recovery time than other agents, and the SUCRA value was 95.2%. (Appendix A).

Overall, lipegfilgrastim 6 mg and eflapegrastim 13.2 mg were the most beneficial agents owing to their high efficacy rates (lipegfilgrastim 6 mg: ANC recovery time 95.2%, FN 97.4%; eflapegrastim 13.2 mg: FN 87%, SN 89.3%) and considerable safety profiles (lipegfilgrastim 6 mg: SAEs 68%; eflapegrastim 13.2 mg: SAEs 67.5%). Shenlida 6 mg was demonstrated to be the safest with the lowest incidence of SAEs. However, it was less effective in DSN and ANC, with the efficacy rates being 50.2% and 30%, respectively. 

#### 3.2.3. Dose-Based Network Meta-Analysis

The dose-based network meta-analysis was conducted to compare the agents with similar chemical structures and different equivalent doses of G-CSF.

We discovered a trend of a dose–effect relationship, and the 3.6 mg, 4.5 mg, 6 mg, and 13.2 mg dosages all performed significantly better than 1.8 mg in reducing DSN (3.6 mg: SMD −0.68 [−1.13, −0.22; moderate]; 4,5 mg: −0,87 [−1.57, −0.17; low]; 6 mg: −0.89 [−1.49, −0.29; moderate]; 13.2 mg: −1.02 [1.63, −0.41; high]). Comparisons between other doses also showed a trend towards a better effect at higher doses, but some results were limited by the wide CrIs.

Notably, compared to the guideline-recommended LA-G-CSF 6 mg, LA-G-CSF 13.2 mg was more effective. Overall, LA-G-CSF 13.2 mg ranked top in DSN and for the incidence of SN and FN (SUCRA, DSN: 91.4%; SN: 96.4%; FN: 95.5%) (Table 2). A dose of 13.2 mg showed a shorter DSN and lower incidence of SN than 6 mg (13.2 mg: DSN, SMD −0.13 [95% CrI, −0.24 to −0.03; low]; SN, RR 0.65 [0.43 to 0.96; very low]) (Figure 5 and Appendix A). However, an increased dosage was associated with a higher risk of SAEs (SUCRA, grade 3–4 SAEs: 30%) (Table 2), although no statistically significant difference was observed in the incidence of SAEs between the two doses (13.2 mg: grade 3–4 SAEs, RR 1.1 [0.45 to 3.0; low]) (Appendix A). 

## 4. Discussion

To the best of our knowledge, this is the first systematic review and network meta-analysis that evaluates the efficacy and safety of LA-G-CSFs for patients with breast cancer, and compares different doses of LA-G-CSFs based on multiple outcome measures. 

G-CSFs are recommended for preventing FN in cancer patients receiving a high-risk (≥20%) chemotherapy regimen, or an intermediate-risk (10%−20%) regimen if there are any risk factors [2,50,51]. Filgrastim and pegfilgrastim are reported to be equally effective in practice guidelines. Additionally, a previous meta-analysis also found no difference between these two drugs [52]. However, LA-G-CSFs are increasingly used compared to SA-G-CSFs [6]. This might be explained by possible underdosage of SA-G-CSFs, and therefore long-acting agents are more effective in reducing the incidence of FN [53]. Based on newly published trials, our network meta-analysis first demonstrated that several LA-G-CSF biosimilars were more effective than SA-G-CSF for patients with breast cancer. Mecapegfilgrastim (100 µg/kg, 6 mg) could effectively reduce the incidence of FN and shorten the DSN. Lipegfilgrastim 6 mg was more effective in reducing the incidence of FN and shortening the ANC recovery time. Empegfilgrastim (7.5 mg, 6 mg) was more effective than SA-G-CSF in reducing the DSN. These results can be attributed to several underlying mechanisms. During hematopoiesis, immature granulocytes are stored in the bone marrow before being gradually released into the blood. SA-G-CSFs selectively induce the release of mature granulocytes, leading to a single peak release 6–8 h after administration, which subsides over 24 h [54]. In contrast, LA-G-CSFs, with their PEG moiety, can stimulate granulocyte production and storage in the bone marrow as well as release mature granulocytes [23], inducing both an early and a late peak release [55]. As a result, LA-G-CSFs can not only enhance granulocyte production but also cause a more substantial, sustained, and uniform release. Our study substantiated that LA-G-CSFs were more effective and safer than SA-G-CSFs. These findings offer compelling evidence to guide the clinical use of LA-G-CSF and its biosimilars.

With the expiration of the patent for Neulasta^®^, multiple novel LA-G-CSF biosimilars have been developed given the substantial economic market for G-CSF agents in alleviating chemotherapy-induced neutropenia. However, direct comparisons of the efficacy of these analogues are lacking. [56,57]. Our study demonstrated the superiority of lipegfilgrastim and eflapegrastim over other LA-G-CSFs, particularly compared to guideline-recommended pegfilgrastim (Neulasta^®^) [51,58,59]. Lipegfilgrastim is quite different from pegfilgrastim in pharmacokinetics and pharmacodynamics [60,61]. The maximum blood concentration of lipegfilgrastim could be reached 30–36 h after administration and the terminal half-life ranges from 32 to 62 h (7–10 h longer compared to the same dose of pegfilgrastim) [57]. Eflapegrastim, a new type of LA-G-CSF, can presumably better reduce drug clearance and promote drug uptake to the bone marrow when coupled to a human IgG4 Fc fragment [48].

In clinical practice, chemotherapy-induced adverse events may occur in patients receiving LA-G-CSF at standard doses. To address this problem, physicians often increase the dose of G-CSFs given to patients with inadequate prophylaxis based on their previous experience, but no direct evidence supports that such practice is beneficial [9,62]. A previous RCT based on 172 women with breast cancer found that an elevated dose of SA-G-CSF (10 mcg/kg/day) did not improve the DSN compared to the original dose (5 mcg/kg/day) (*p* = 0.93) [63]. In contrast, our study pioneeringly demonstrated that increasing the dose of LA-G-CSF could reduce the incidence of SN and FN, and shorten the time of ANC recovery. Furthermore 3.6 mg, 4.5 mg, 6 mg, and 13.2 mg dosages performed significantly better than 1.8 mg in reducing the DSN. A trend of a dose–effect relationship was observed in this study. The wide CrIs in the network meta-analysis could be explained by the differences in the sample size and endpoints among the included RCTs. More well-designed and large-scale studies on G-CSFs for chemotherapy-induced neutropenia are needed. 

Furthermore, the association between high-dose LA-G-CSF and an increased risk of SAEs remains to be elucidated. Our study observed no significant increase in the incidence of SAEs during LA-G-CSF administration at an escalated dose. Dufour et al. [64] presented a case report of a 2-year-old child (administered a dose 9.37-fold higher than scheduled) and a 79-year-old adult (given a dose 8-fold higher than reference), and they discovered that an overdose of pegfilgrastim may cause an uneventful outcome in children and produce manageable side effects in older patients. Overall, multiple factors, such as efficacy and safety, should be considered when deciding whether to increase the dose of LA-G-CSF, and its dose should be adjusted in a timely manner according to actual conditions. Our study boasts multiple advantages. In the present study, additional Chinese databases were searched to evaluate novel LA-G-CSFs. We included several new LA-G-CSFs (Shenlida, Jin You Li, Xinruibai) that are widely used in Chinese healthcare institutions but rarely analyzed. Our study proved for the first time that patients receiving Shenlida were the least prone to developing grade 3–4 AEs than those receiving other LA-G-CSFs. A previous meta-analysis pooled the estimates of different doses of the same drug without considering individual doses, which could lead to inconsistent results. With a tailored study design (drug- and dose-based network meta-analysis), we are the first to investigate the feasibility of LA-G-CSF dose escalation and highlight the superiority of particular LA-G-CSFs in specific domains. Our results will aid clinicians in further managing life-threatening neutropenia induced by chemotherapy, giving precise and effective G-CSF medications, and developing tailored drug and dosage regimens for individual patients. For patients who suffer from severe neutropenia even under a standard LA-G-CSF regimen, an elevated dose of LA-G-CSF could be considered to reduce the impact of neutropenia. Furthermore, a potential dose–effect correlation in LA-G-CSF is observed in our study, which lays the groundwork for subsequent investigations on devising innovative LA-G-CSF protocols. It reveals that alternative dosage strategies may provide enhanced therapeutic effects with lower costs.

However, this study also has some limitations. First, since none of the interventions had more than 10 head-to-head inter-comparison trials, a publication bias assessment was unfeasible. Second, in the dose-based analysis, the therapeutic effect of certain doses may have been biased due to the neglect of subtle structural differences among different manufacturers and suboptimal drug performances. Third, we provided SUCRA to estimate the ranking probability of the effects of different agents and doses, but the results should be interpreted with caution. To better interpret the data, we have presented the results of the network analysis of each comparison as SMDs and RRs. Finally, different chemotherapy regimens may induce complications of different degrees, which could lead to an inevitable bias on the efficacy and safety of G-CSFs.

## 5. Conclusions

In this systemic review and network meta-analysis, we found that several LA-G-CSFs were more effective and safer than both SA-G-CSF and original pegfilgrastim for improving the chemotherapy outcomes in breast cancer patients A trend of a dose–effect relationship in LA-G-CSF was discovered, and an increased dosage of LA-G-CSF might enhance efficacy without additional SAEs. More RCTs are needed to further assess the efficacy and safety of higher-dose LA-G-CSF therapy. Prophylactic use of LA-G-CSF in patients with breast cancer should be based on the pharmacological activities of individual agents and the patients’ conditions.

## Figures and Tables

**Figure 1 cancers-15-03675-f001:**
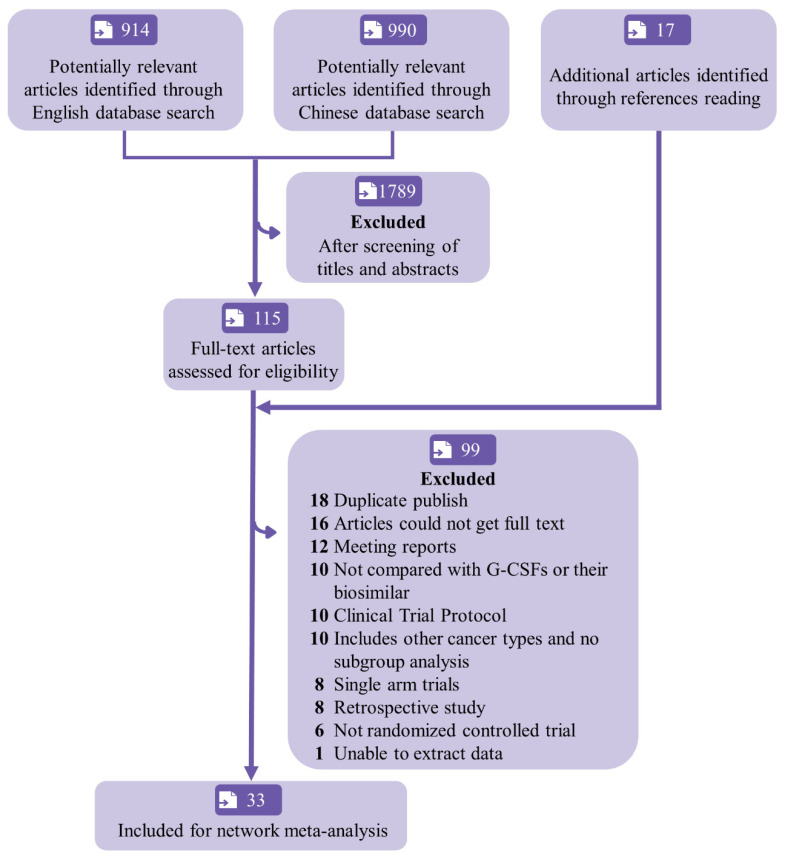
Flowchart of study selection and design. G-CSF = granulocyte colony-stimulating factor.

**Figure 2 cancers-15-03675-f002:**
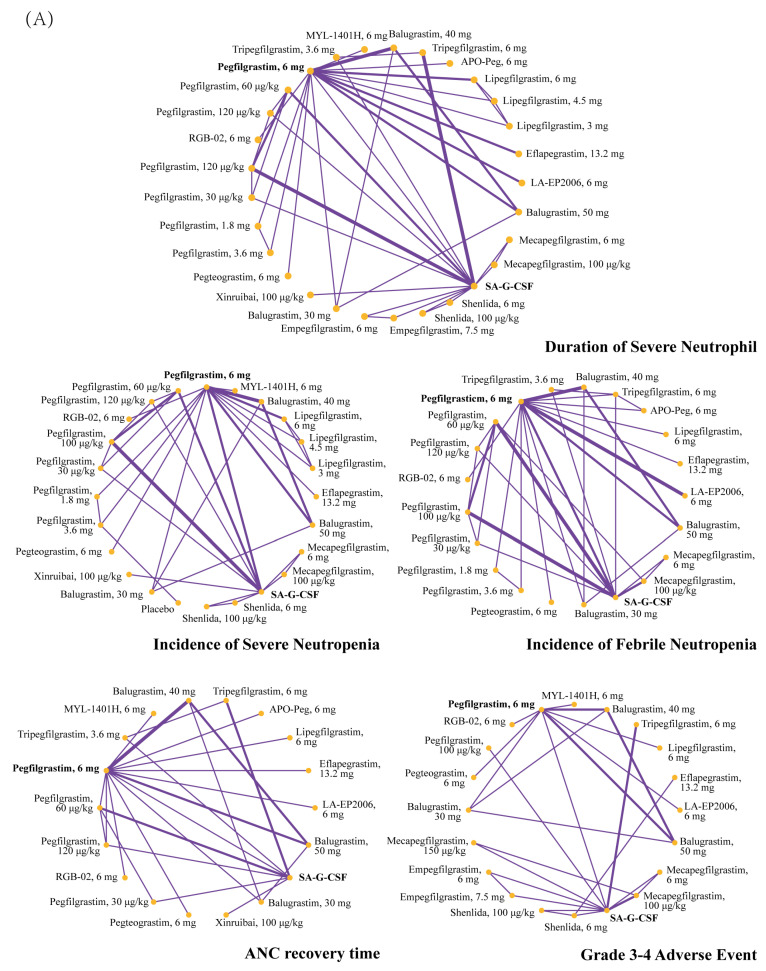
Network graph for evaluated outcomes in dose-based analysis (**A**) and drug-based analysis (**B**). Line width is proportional to the number of RCTs comparing each pair of treatments. ANC = absolute neutrophil count. SA-G-CSF = short-acting G-CSF.

**Figure 3 cancers-15-03675-f003:**
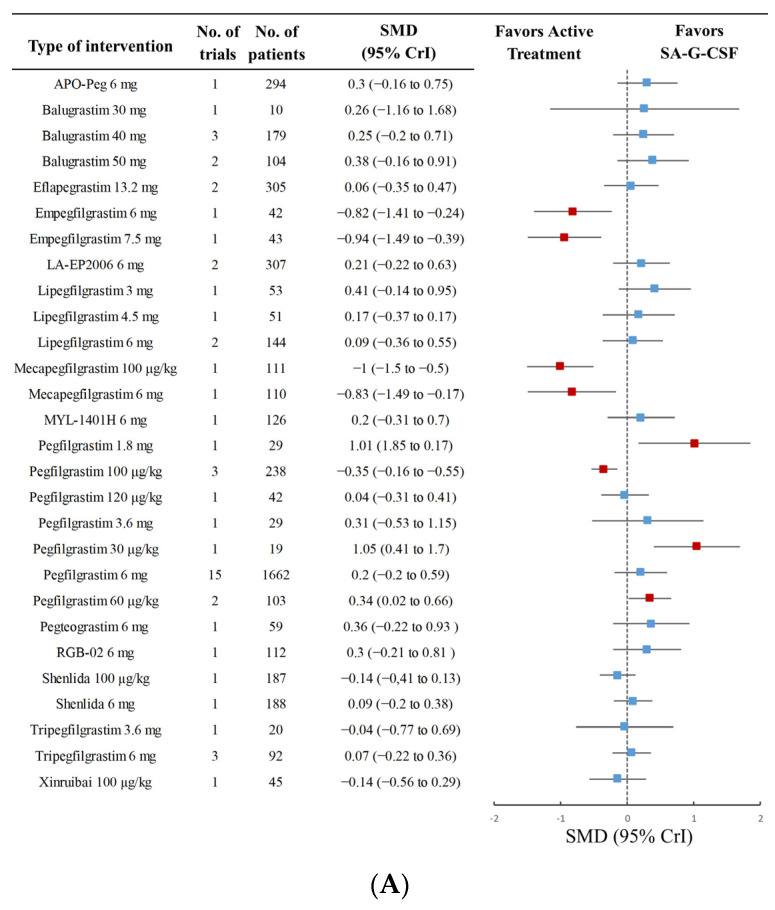
Forest plots of the network meta-analysis for DSN (**A**) and the incidence of FN (**B**) compared to SA-G-CSF. Two studies were removed and a sensitivity analysis for incidence of FN (**C**) was conducted. The bars indicate 95% credibility intervals (CrIs). Red squares represent 95% Crl with statistical significance; Blue ones are no statistically significant. RR = risk ratio, SMD = standardized mean difference, DSN = duration of severe neutropenia, FN = febrile neutropenia, SA-G-CSF = short-acting granulocyte colony-stimulating factor.

**Figure 4 cancers-15-03675-f004:**
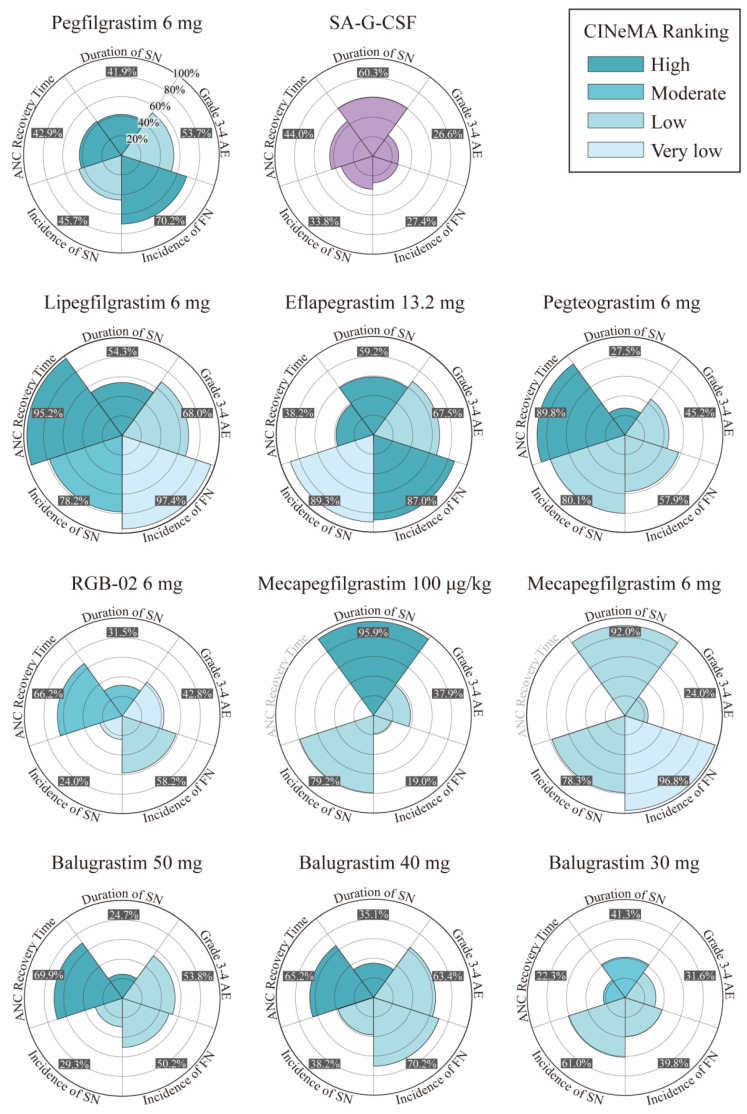
Summary of rose charts for the overall profile of most treatments across the five outcomes. Darker hues of cyan indicate higher levels of confidence in the evidence for the comparison of the corresponding treatments vs. SA-G-CSF (categorized into four levels: very low, low, moderate, and high quality). The light purple color identifies SA-G-CSF as the common comparator. SA-G-CSF = short-acting granulocyte colony-stimulating factor.

**Figure 5 cancers-15-03675-f005:**
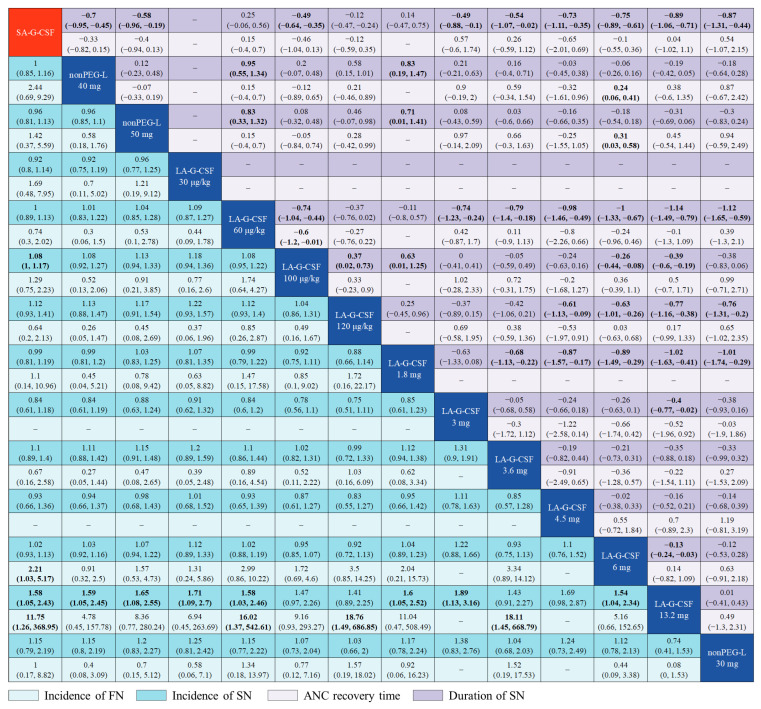
Network meta-analysis for ANC recovery time, DSN, and the incidence of FN and SN in does-based analysis. SMDs with 95% CrI are presented for ANC recovery time and duration of SN in the upper right section of the table, while RRs with 95% CrIs are presented for incidence of FN and SN in the lower left part of the table. For ANC recovery time and duration of SN, SMDs below 0 favor the treatment defined by the column. For incidence of FN and SN, RRs below 1 favor the treatment defined by the row. Bold results indicate statistical significance. SA-G-CSF = short-acting granulocyte colony-stimulating factor, ANC = absolute neutrophil count, DSN = duration of severe neutropenia, FN = febrile neutropenia, SN = severe neutropenia, SMD = standardized mean difference, RR = risk ratio, non-PEG-L = non-pegylated LA-G-CSF.

**Table 1 cancers-15-03675-t001:** Basic characteristic of the included studies.

No.	Study ID	Region	Type of Intervention and Dose	Initiation Time of the G-CSF	Number of Patients	Age (Mean, SD)	Chemotherapy Regimen
1	Holmes et al., study 1, 2002 [21]	USA	SA-G-CSF	24 h post-chemotherapy and continued (1) for 14 days, or (2) until the ANC reached 10 × 10^9^/L post-ANC nadir.	25	50, 9	AT (60/75, 4 cycles)
			Pegfilgrastim, 30 μg/kg	24 h post-chemotherapy	19	51, 13	
			Pegfilgrastim, 60 μg/kg	24 h post-chemotherapy	60	51, 11	
			Pegfilgrastim, 100 μg/kg	24 h post-chemotherapy	46	49, 11	
2	Holmes et al., study 2, 2002 [22]	Multi	SA-G-CSF	24 h post-chemotherapy and continued (1) for 14 days, or (2) until the ANC reached 10 × 10^9^/L post-ANC nadir.	149	50.9, 11.7	AT (60/75, 4 cycles)
			Pegfilgrastim, 100 μg/kg	24 h post-chemotherapy	147	51.9, 11.1	
3	Green et al., 2003 [23]	Multi	SA-G-CSF	≥24 h post-chemotherapy, continued daily until (1) an ANC ≥ 10.0 × 10^9^/L was documented after the expected nadir or (2) for a maximum of 14 days.	75	52.8, 11.5	AT (60/75, 4 cycles)
			Pegfilgrastim, 6 mg	24 h post-chemotherapy	77	52.1, 9.2	
4	Vogel et al., 2005 [24]	Multi	Placebo	24 h post-chemotherapy	465	52.1, 8.7	T (100, 4 cycles)
			Pegfilgrastim, 6 mg	24 h post-chemotherapy	463	51.9, 11.2	
5	Minckwitz et al., 2008 [25]	German	filgrastim 5 ug/kg/day or lenograstim 150 ug/m^2^/day	24 h post-chemotherapy	377	-	TAC (75/50/500, 6 cycles)
			pegfilgrastim 6.0 mg	24 h post-chemotherapy	305	-	
6	Bondarenko et al., 2013 [26]	Multi	Lipegfilgrastim, 6 mg	24 h post-chemotherapy	94	49.3, 10.0	AT (60/75, 4 cycles)
			Pegfilgrastim, 6 mg	24 h post-chemotherapy	94	50.9, 9.3	
7	Park et al., 2013 [27]	Korea	SA-G-CSF	24 h after chemotherapy and continued until (1) documented ANC 5 × 10^9^/L after nadir, or (2) up to 10 days.	21	45.29, 6.13	TAC (70/75/500, 6 cycles)
			DA-3031, 3.6 mg	24 h post-chemotherapy	20	42.50, 5.62	
			DA-3031, 6 mg	24 h post-chemotherapy	20	46.95, 9.19	
8	Volovat et al., 2014 [29]	Multi	Pegfilgrastim, 6 mg	24 h post-chemotherapy	151	50.8, 9.65	AT (60/75, 4 cycles)
			Balugrastim, 40 mg	24 h post-chemotherapy	153	51.5, 10.28	
9	Buchner et al., 2014 [28]	Multi	Pegfilgrastim, 6 mg	24 h post-chemotherapy	54	49.5, 11.1	AT (60/75, 4 cycles)
			Lipegfilgrastim, 3 mg	24 h post-chemotherapy	53	53.1, 9.2	
			Lipegfilgrastim, 4.5 mg	24 h post-chemotherapy	51	52.8, 10.1	
			Lipegfilgrastim, 6 mg	24 h post-chemotherapy	50	51.4, 9.8	
10	Filon et al., 2015 [30]	Russia	SA-G-CSF	24 h post-chemotherapy, until (1) ANC reached 10,000/μL or (2) for 14 days.	43	51.86, 9.68	AT (60/75, 4 cycles)
			Empegfilgrastim, 6 mg	24 h post-chemotherapy	42	50.42, 9.19	
			Empegfilgrastim, 7.5 mg	24 h post-chemotherapy	43	48.38, 1.07	
11	Gladkov et al., 2015 [31]	Multi	Pegfilgrastim, 6 mg	24 h post-chemotherapy	25	52.8, 10.4	AT (60/75, 4 cycles)
			Balugrastim, 30 mg	24 h post-chemotherapy	10	56.9, 9.5	
			Balugrastim, 40 mg	24 h post-chemotherapy	21	51.4, 10.3	
			Balugrastim, 50 mg	24 h post-chemotherapy	20	53.8, 9.5	
12	Kosaka et al., 2015 [32]	Japan	Placebo, 3.6 mg	24 h post-chemotherapy	173	49.8, 7.8	TC (75/600, 4–6 cycles)
			Pegfilgrastim, 3.6 mg	24 h post-chemotherapy	173	50.7, 8.0	
13	Lee et al., 2016 [38]	South Korea	Pegteograstim 6.0 mg	24 h post-chemotherapy	57	49,10.5	TAC (75/50/500, 6 cycles)
			Pegfilgrastim 6.0 mg	24 h post-chemotherapy	59	49,11	
14	Masuda et al., 2015 [33]	Japan	Pegfilgrastim, 1.8 mg	24 h post-chemotherapy	29	46.5, 7.4	TAC (70/75/500, 6 cycles)
			Pegfilgrastim, 3.6 mg	24 h post-chemotherapy	29	46.5, 9.4	
			Pegfilgrastim, 6.0 mg	24 h post-chemotherapy	29	47.3, 9.4	
15	Zhang et al., 2015 [34]	China	SA-G-CSF	48 h post-chemotherapy, continued either for 14 days or until the ANC reached 10^9^/L post-ANC nadir, whichever occurred first, but for at least 7 days.	43	47.35, 8.14	TAC (70/75/500, 6 cycles)
			Pegfilgrastim, 60 μg/kg	48 h post-chemotherapy	43	47.03, 7.66	
			Pegfilgrastim, 100 μg/kg	48 h post-chemotherapy	43	48.1, 8.09	
			Pegfilgrastim, 120 μg/kg	48 h post-chemotherapy	42	46.71, 6.80	
16	Blackwell et al., 2016 [35]	Multi	Pegfilgrastim, 6.0 mg	24 h post-chemotherapy	153	49.1, 10.07	TAC (70/75/500, 6 cycles or more)
			LA-EP2006, 6.0 mg	24 h post-chemotherapy	155	48.8, 10.5	
17	Gladkov et al., 2016 [36]	Multi	Pegfilgrastim, 6 mg	24 h post-chemotherapy	86	50.3, 9.1	AT (60/75, 4 cycles)
			Balugrastim, 40 mg	24 h post-chemotherapy	85	49.2, 9.9	
			Balugrastim, 50 mg	24 h post-chemotherapy	84	49.8, 9.6	
18	Harbeck et al., 2016 [37]	Multi	LA-EP2006, 6 mg	24 h post-chemotherapy	159	49.9, 9.5	TAC (70/75/500, 6 cycles)
			Pegfilgrastim, 6 mg	24 h post-chemotherapy	157	50.5, 10.9	
19	Harbeck et al., 2017 [39]	Multi	LA-EP2006, 6 mg	24 h post-chemotherapy	90	47.8, 10.42	TAC (70/75/500, 6 cycles)
			Pegfilgrastim, 6 mg	24 h post-chemotherapy	84	47.4, 9.53	
20	Park et al., 2017 [40]	Korean	SA-G-CSF	24 h post-chemotherapy and continued until (1) ANC was documented to be 5 × 10^9^/L after nadir, or (2) for up to 10 days	38	45.76, 8.12	TAC (75/50/500, 6 cycles)
			DA-3031, 6 mg	24 h post-chemotherapy	36	47.11, 6.37	
21	Ashrafi et al., 2018 [42]	Iran	SA-G-CSF	24 h post-chemotherapy	12	45.3, 10.5	AC-T (60/600/80)
			Pegfilgrastim, 6 mg	24 h post-chemotherapy	12	45.3, 10.5	
22	Desai et al., 2018 [43]	Multi	APO-Peg, 6 mg	24 h post-chemotherapy	294	51.9, 10.0	TAC (75/50/500, 6 cycles)
			Pegfilgrastim, 6 mg	24 h post-chemotherapy	295	51.4, 10.3	
23	Huang et al., 2018 [44]	China	SA-G-CSF	48 h post-chemotherapy, continued for 14 days or until the ANC became ≥ 10 × 10^9^/L.	22	47.5, 7	AC (60/600, 4 cycles)
			Jin You Li, 100 μg/kg	48 h post-chemotherapy	24	46.5, 6.75	
24	Wu et al., 2018 [41]	China	SA-G-CSF	48 h post-chemotherapy, continued until (1) a documented ANC ≥ 5.0×10^9^/L twice post-ANC nadir; (2) for up to 14 days.	44	59.25, 6.79	EC-T (100/600/100, 4 cycles)
			Xinruibai, 100 μg/kg	48 h post-chemotherapy	45	52.36, 5.48	
25	Xie et al., 2018 [11]	China	SA-G-CSF	48 h post-chemotherapy, and continued daily until (1) an ANC ≥ 5.0 × 10^9^/L or (2) for a maximum of 14 days.	194	49.22, 9.24	EC (100/600, 4 cycles) or TC (75/75, 4 cycles) or ET (600/75, 4 cycles)
			Shenlida, 100 µg/kg	48 h post-chemotherapy	187	47.12, 8.81	
			Shenlida, 6 mg	48 h post-chemotherapy	188	49.40, 8.84	
26	Kahan et al., 2019 [45]	Multi	RGB-02, 6 mg	24 h post-chemotherapy	121	51.0, 8.20	AT (60/75, 4 cycles)
			Pegfilgrastim, 6 mg	24 h post-chemotherapy	118	51.2, 9.56	
27	Sohn et al., 2019 [49]	Korea	SA-G-CSF	On D3, D5, D7, D9, and D11 of each cycle	11	55, 3.8	TAC (75/50/500, 6 cycles)
			Pegfilgrastim, 6 mg	24 h post-chemotherapy	11	50.6, 10.0	
28	Waller et al., 2019 [46]	Multi	MYL-1401H, 6 mg	24 h post-chemotherapy	127	50, 11	TAC (75/50/500, 6 cycles)
			Pegfilgrastim, 6 mg	24 h post-chemotherapy	67	50, 10	
29	Wang et al., 2019 [10]	China	SA-G-CSF	48 h post-chemotherapy and continued (1) for 14 days; (2) until ANC ≥ 5.0 × 10^9^/L in two consecutive examinations after ANC reached the lowest point; (3) until ANC ≥ 15×10^9^/L.	61	47.84, 8.67	AT (75/75, 4 cycles) or AC (100/600, 4 cycles)
			HHPG-19K, 100 μg/kg	48 h post-chemotherapy	60	47.58, 8.88	
			HHPG-19, 150 μg/kg	48 h post-chemotherapy	61	48.97, 8.59	
30	Xu et al., 2019 [12]	China	SA-G-CSF	48 h post-chemotherapy, continued (1) until a documented ANC ≥ 5.0×10^9^/L twice; (2) until ANC ≥ 15×10^9^/L once after the expected nadir; (3) for up to 14 days.	111	48.21, 8.55	AT (75/75, 4 cycles) or AC (100/600, 4 cycles)
			Mecapegfilgrastim (HHPG-19K), 6 mg	48 h post-chemotherapy	110	47.37, 8.60	
			Mecapegfilgrastim (HHPG-19K), 100 μg/kg	48 h post-chemotherapy	110	48.03, 9.01	
31	Cobb et al., 2020 [47]	Multi	Pegfilgrastim, 6 mg	24 h post-chemotherapy	119	59.2, 10.5	TC (75/600, 4 cycles)
			Eflapegrastim, 13.2 mg	24 h post-chemotherapy	118	57.6, 10	
32	Schwartzberg et al., 2020 [48]	USA	Pegfilgrastim, 6 mg	24 h post-chemotherapy	210	59.59, 10.91	TC (75/600, 4 cycles)
			EflaPegrastim, 13.2 mg	24 h post-chemotherapy	196	60.61, 10.08	
33	Liang et al., 2021 [13]	China	SA-G-CSF	24 h post-chemotherapy and continued for 3 days	20	64.79, 6.34	not clear
			Jin You Li, 6 mg	24 h post-chemotherapy	20	64.45, 5.87	

ANC = absolute neutrophil count; TAC = docetaxel + doxorubicin + cyclophosphamide; T = docetaxel; TC = docetaxel + cyclophosphamide; AC = doxorubicin + cyclophosphamide; AT = doxorubicin + docetaxel; AC-T = doxorubicin + cyclophosphamide + paclitaxel; EC-T = pharmorubicin + cyclophosphamide + docetaxel; EC = pharmorubicin + cyclophosphamide; SA-G-CSF = short-acting granulocyte colony-stimulating factor.

**Table 2 cancers-15-03675-t002:** SUCRA results (percentages) for each dose of LA-G-CSF biosimilars, nonPEG-LA-CSFs, and an SA-G-CSF.

Dose	Outcome				
	DSN	Incidence of SN	Incidence of FN	ANC Recovery Time	Grade 3–4 AE
SA-G-CSF	14.6%	42.4%	38.7%	41.4%	49.7%
LA-G-CSF biosimilar 6 mg	74.3%	53.0%	74.9%	48.4%	44.5%
LA-G-CSF biosimilar 13.2 mg	91.4%	96.4%	95.5%	42.8%	30.0%
LA-G-CSF biosimilar 7.5 mg	66.9%	-	-	-	89.1%
LA-G-CSF biosimilar 4.5 mg	71.1%	33.6%	-	78.8%	-
LA-G-CSF biosimilar 3.6 mg	52.5%	70.4%	25.9%	29.0%	-
LA-G-CSF biosimilar 3 mg	45.4%	15.2%	-	17.7%	-
LA-G-CSF biosimilar 1.8 mg	10.2%	39.5%	46.0%	-	-
LA-G-CSF biosimilar 150 μg/kg	-	-	31.1%		-
LA-G-CSF biosimilar 120 μg/kg	21.2%	72.7%	23.5%	54.7%	-
LA-G-CSF biosimilar 100 μg/kg	43.6%	69.5%	52.9%	79.4%	80.5%
LA-G-CSF biosimilar 60 μg/kg	3.6%	43.1%	27.8%	31.8%	-
LA-G-CSF biosimilar 30 μg/kg	-	23.0%	62.0%	-	-
non-PEG-LA-CSF 50 mg	53.7%	30.5%	53.6%	78.9%	47.9%
non-PEG-LA-CSF 40 mg	67.2%	42.7%	76.1%	72.8%	62.8%
non-PEG-LA-CSF 30 mg	84.3%	68.0%	41.8%	24.3%	15.5%

SUCRA: surface under the cumulative ranking curve. SUCRA values can range from 0% (i.e., the treatment always ranks last) to 100% (i.e., the treatment always ranks first). SA-G-CSF = short-acting granulocyte colony-stimulating factor, DSN = duration of severe neutropenia, FN = febrile neutropenia, SN = severe neutropenia, ANC = absolute neutrophil count, AE = adverse event.

## Data Availability

The data presented in this study are available on request from the corresponding author.

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
