# Peer review of "Assessing the Optimal Regimen: A Systematic Review and Network Meta-Analysis of the Efficacy and Safety of Long-Acting Granulocyte Colony-Stimulating Factors in Patients with Breast Cancer"

_cancers, 2023, doi:10.3390/cancers15143675_

Round 1

Reviewer 1 Report

The study has been well prepared, performed and presented. The only remark is that it would be useful for clinicians to show what grade 3-4 side effects were observed during the use of particular drugs, did they differ depending on the drug or the dose?

Reviewer 2 Report

29-30 a description of the PICO is missing in the methods, furthermore at least a draft of the selection is missing in the results. in the abstract you can omit the software and the package.. mostly insert the type of frequentist or Bayesian model..

34 softened as in simple summary "seems/appears..", didn't you conduct a RoB-2?

89 enter the registration in the abstract.

99 inclusion criteria cannot be omitted, in fact the PICO is missing. RCTs only? language restriction?

111 there is the RoB-2

183 I would suggest including a description of the Netplots such as "the size of the circle represents the number of patients included, the thickness of the line expressing the number of comparisons between interventions" Reference to: https://pubmed.ncbi.nlm.nih.gov/32478581/

234 also here: I would suggest describing the representation more as "the graphical figure pairwise comparisons of each network versus control intervention" Reference to: https://doi.org/10.1016/j.rehab.2021.101602

Figure 5 represents the ranking table, it should be considered immediately after the netplots, because it includes both pairwise and network comparisons. furthermore at least this should be described in the methods: “Thus in a ranking table, we will report above the leading diagonal the pairwise effect size comparisons, below the leading diagonal ESs estimated from network meta-analyses” Reference to: https://pubmed.ncbi .nlm.nih.gov/32478581/

Author Response

Please see the attachment, thanks!

Reviewer 3 Report

Comments: The manuscript by Zhixuan You et al. titled “Assessing the Optimal Regimen: A Systematic Review and Network Meta-Analysis on the Efficacy and Safety of Long-Acting Granulocyte Colony-Stimulating Factors in Patients with Breast Cancer” provides nice work about the usage of LA-G-CSF. There are, however, some questions and suggestions as follow:   1. What is originality of the review compring with PMID: 25284721?   2. Why this author chose breast cancer to assess the optimal regimen?   3. All the relevant studies in this review are related to chemotherapy. How about the usage of G-CSF after radiotherapy?   4. Does the initiation time of the G-CSF affect the efficacy and safety?   5. In Table 1, the Age column of No. 21 and No. 22 should be presented like the style of others.   6. Figure 2 should add the “A” and “B” panels within the figure.   7. Figure 5, the p values should be shown in the figure.   8. The figure legends should be elaborated to explain the details, e.g. it is unclear why some of the texts are in bold in figure 2.   9. Some of the elements in the figures are too small and blur, which should be improved.  

Typos and unfriendly mode of English usage are found.

Author Response

Please see the attachment. We greatly appreciate your time and valuable insights. Thank you!

Round 2

Reviewer 2 Report

Dear authors, I can consider your answers to my concerns satisfactory. I suggest the suitability of your manuscript for publication